# A Comprehensive Overview of Intraoperative Complications during Retzius-Sparing Robot-Assisted Radical Prostatectomy: Single Series from High-Volume Center

**DOI:** 10.3390/cancers16071385

**Published:** 2024-03-31

**Authors:** Alberto Olivero, Stefano Tappero, Francesco Chierigo, Ofir Maltzman, Silvia Secco, Erika Palagonia, Antonio Piccione, Aldo Massimo Bocciardi, Antonio Galfano, Paolo Dell’Oglio

**Affiliations:** 1Department of Urology, ASST Grande Ospedale Metropolitano Niguarda, 20162 Milan, Italy; oliveroalby@gmail.com (A.O.); stefano.tappero@ospedaleniguarda.it (S.T.); francesco.chierigo@ospedaleniguarda.it (F.C.); ofir.malztman@ospedaleniguarda.it (O.M.); silvia.secco@ospedaleniguarda.it (S.S.); erika.palagonia@ospedaleniguarda.it (E.P.); antonio.piccione@ospedaleniguarda.it (A.P.); aldo.bocciardi@ospedaleniguarda.it (A.M.B.); antonio.galfano@ospedaleniguarda.it (A.G.); 2Department of Urology, IRCCS Ospedale Policlinico San Martino, University of Genova, 16126 Genova, Italy; 3Department of Surgical and Diagnostic Integrated Sciences (DISC), University of Genova, 16126 Genova, Italy; 4Department of Urology, Netherlands Cancer Institute-Antoni Van Leeuwenhoek Hospital, 1066 CX Amsterdam, The Netherlands; 5Interventional Molecular Imaging Laboratory, Department of Radiology, Leiden University Medical Center, 2333 ZA Leiden, The Netherlands

**Keywords:** prostate cancer, robot-assisted radical prostatectomy, adverse events, intraoperative complications, robotic surgery

## Abstract

**Simple Summary:**

Intraoperative complications during Retzius-sparing robot-assisted radical prostatectomy are uncommon, affecting 2.1% of patients in our series of almost two thousand cases. Only 0.37% of the patients treated with Retzius-sparing robot-assisted radical prostatectomy developed post-operative sequelae due to Intraoperative complications. When divided according to the time of the procedure 42% of the events were ePNLD-related, and the 58% were RARP-related Intraoperative complications are more likely to occur in older individuals with a higher BMI. Patient suffering intraoperative complications demonstrated a higher rate of intraoperative blood transfusion, and a longer length of stay. These complications are mostly of low severity but underline the need for careful monitoring and reporting.

**Abstract:**

Background: Intraoperative complications (ICs) are invariably underreported in urological surgery despite the recent endorsement of new classification systems. We aimed to provide a detailed overview of ICs during Retzius-sparing robot-assisted radical prostatectomy (RS-RARP). Methods: We prospectively collected data from 1891 patients who underwent RS-RARP at a single high-volume European center from January 2010 to December 2022. ICs were collected based on surgery reports and categorized according to the Intraoperative Adverse Incident Classification (EAUiaiC). The quality criteria for accurate and comprehensive reporting of intraoperative adverse events proposed by the Intraoperative Complications Assessment and Reporting with Universal Standards (ICARUS) Global Surgical Collaboration Project were fulfilled. To better classify the role of the RS-RARP approach, ICs were classified into anesthesiologic and surgical ICs. Surgical ICs were further divided according to the timing of the complication in RARP-related ICs and ePNLD-related ICs. Results: Overall, 40 ICs were reported in 40 patients (2.1%). Ten out of thirteen ICARUS criteria were satisfied. According to EAUiaiC grading of ICs, 27 (67.5%), 7 (17.5%), 2 (5%), 2 (5%), and 2 (5%) patients experienced Grade 1, 2, 3, 4A, and 4B, respectively. When we classified the ICs, two cases (5%) were classified as anesthesiologic ICs. Among the 38 surgical ICs, 16 (42%) were ePNLD-related, and 22 (58%) were RARP-related. ICs led to seven (0.37%) post-operative sequelae (four non-permanent and three permanent). Patients who suffered ICs were significantly older (67 years vs. 65 years, *p* = 0.02) and had a higher median BMI (27.0 vs. 26.1, *p* = 0.01), but did not differ in terms of comorbidities or tumor characteristics (all *p* values ≥ 0.05). Conclusions: Intraoperative complications during RS-RARP are relatively infrequent, but should not be underestimated. Patients suffering from ICs are older, have a higher body mass index, a higher rate of intraoperative blood transfusion, and a longer length of stay.

## 1. Introduction

Prostate cancer (PCa) affects over one million men annually and is the most frequent non-cutaneous malignancy in males. In the last 20 years, robot-assisted radical prostatectomy (RARP) has been recognized as one of the standard surgical treatments and has rapidly become the most common pelvic surgery [1,2].

Retzius-sparing robot-assisted radical prostatectomy (RS-RARP), after being proposed more than a decade ago [3], has rapidly gained success worldwide [4] due to clinically improved continence recovery relative to the standard approach, as demonstrated in randomized clinical trials and metanalyses [5,6,7]. Nevertheless, several authors claim that RS-RARP is a complex technique with a prolonged learning curve that could lead to adverse events [8].

RARP, in general, is not devoid of intraoperative complications (ICs), as with all major surgical procedures. Albeit relatively infrequent, these complications may significantly affect the patient’s postoperative course [9,10], as well as the surgeon’s psychological well-being [11]. ICs management represents a paramount component of surgical practice; thus, a proper understanding of these events is crucial for improving the safety and efficacy of any surgical procedure, and RARP does not stand as an exception. Although flourishing research is available regarding the rates, classification, and prediction scores of postoperative complications after RARP [12,13,14], ICs are often under-reported [15]. Consequently, despite the recent endorsement of new classification systems, the current body of literature only invariably relies on ballpark estimates of ICs during RARP, especially in the specific setting of RS-RARP [16,17]. 

Given such a knowledge gap, the current study aimed to address the rate and provide a punctual description and classification of ICs during RS-RARP. We relied on the data of patients treated with RS-RARP at a high-volume European center between 2010 and 2022.

## 2. Materials and Methods

### 2.1. Study Population 

This is a retrospective study based on our prospectively collected database including all the patients who underwent RS-RARP at ASST Grande Ospedale Metropolitano Niguarda (Milan, Italy) from January 2010 to December 2022. Each patient was preoperatively counseled, as the contemporary European Association of Urology (EAU) guidelines Campo [1] recommended. Extended pelvic lymph node dissection (ePLND) has been performed in intermediate- and high-risk diseases since 2019, when the risk of lymph node involvement was >7%, according to Gandaglia’s nomogram [18].

All RS-RARP procedures were performed with a four-arm da Vinci Surgical System; the SI System was used until December 2021, and the Xi System was introduced in January 2022 (Intuitive Surgical, Sunnyvale, CA, USA). As previously described, RS-RARP was conducted with a transperitoneal approach by ten robotic surgeons with different degrees of experience. The indication for mono- or bilateral neuro-vascular bundle preservation (nerve-sparing [NS]) was based on the oncological characteristics at diagnosis and the treating physician’s clinical judgment regardless of preoperative erectile function. No patient in the present study received pelvic radiotherapy before the prostatectomy.

### 2.2. Variables of Interest and Study Endpoints

All patients had complete clinical preoperative and intraoperative data, consisting of age, body mass index (BMI), prostate-specific antigen (PSA) at diagnosis, biopsy Gleason score, clinical stage, and previous abdominal or benign prostate hyperplasia (BPH) surgery. Comorbidities were reported according to the Charlson Comorbidity Index (CCI), and were not age adjusted.

The endpoint of this study was to determine the rate of ICs and to describe and classify ICs that occurred during RS-RARP. ICs were defined and categorized according to the Intraoperative Adverse Incident Classification (EAUiaiC) proposed by the European Association of Urology ad hoc Complications Guidelines Panel. The quality criteria for accurate and comprehensive reporting of intraoperative adverse events proposed by the Intraoperative Complications Assessment and Reporting with Universal Standards (ICARUS) Global Surgical Collaboration Project were fulfilled [16,17]. Complications were retrieved by two different urologists (AO and OM), and any discordance was judged and solved by a third expert physician (PDO). Furthermore, to better classify the role of the RS-RARP approach, ICs were classified into anesthesiologic and surgical ICs. Surgical ICs were divided according to the timing of the complication in RARP-related ICs and PNLD-related ICs. 

### 2.3. Surgical Technique

A four-arm da Vinci Si or XI Surgical System (Intuitive Surgical, Sunnyvale, CA, USA) was used in all the cases. All prostatectomies were performed at least one month after prostate biopsy by ten different surgeons. 

The surgical procedure involves placing the patient in the Trendelenburg position at a 30-degree angle. Pneumoperitoneum is induced either by the Veress needle for patients without prior abdominal surgery or by an open laparoscopy incision for others. Six laparoscopic trocars are inserted. The da Vinci robot is configured with the lens facing downward at a 30-degree angle during seminal vesicle isolation and upward during subsequent steps.

The parietal peritoneum is incised for 5–7 cm at the anterior surface of the pouch of Douglas, and seminal vesicles and vasa deferentia are isolated and incised. Denonvillier fascia is separated in an antegrade direction along the posterolateral surface of the prostate, reaching the prostatic apex, maintaining a 360-degree interfacial plane whenever feasible and oncologically safe. In cases of adherences, palpable disease, or doubts, the surgeon opts for a broader dissection plane, following the concept of an incremental nerve-sparing procedure. The vesicoprostatic junction is identified, isolated, and sectioned, sparing the bladder neck when oncologically safe. Two short stitches are placed at the 6 and 12 o’clock positions to identify the bladder neck orifice during the initial phase of the anastomosis. The anterior surface of the prostate is bluntly isolated from the Santorini plexus without cutting. Apex isolation is completed, and the urethra is incised. The prostate is positioned into an endo-bag. The anastomosis is performed using a modified Van Velthoven technique with two needles starting from the 12 o’clock position.

According to the guidelines of the period, external and internal iliac, obturator, and occasionally common iliac lymph node dissections are performed. This involves extending the initial peritoneal incision into a U-shaped one, terminating in the space between the umbilical artery and the internal inguinal ring.

### 2.4. Statistical Analysis 

The statistical analysis was performed using IBM SPSS (Version 24, IBM SPSS Inc., Chicago, IL, USA). Data were summarized using the median and interquartile range (IQR) and absolute and relative frequencies for continuous and categorical variables. Mann–Whitney and Pearson chi-squared tests were used for comparison where appropriate. Patient characteristics and surgical parameters were compared according to the presence or absence of ICs. All tests were two-sided, with a significance level set at *p* < 0.05.

## 3. Results

Of the 1933 patients identified from the database, 42 were excluded due to lack of information: 39 missing baseline characteristics and 3 missing intraoperative reporting; thus, data from 1891 patients were examined in the final analysis. 

Overall, 40 ICs were reported in 40 patients (2.2%). Ten out of thirteen ICARUS criteria were satisfied (Appendix A). According to EAUiaiC, 27 patients (67.5%) experienced Grade 1 ICs, 7 (17.5%) experienced Grade 2 ICs, 2 (5%) experienced Grade 3 ICs, 2 (5%) experienced Grade 4A ICs, and 2 (5%) experienced Grade 4B ICs. The complete list of ICs and their grades is reported in Table 1. Five (12.5%) complications occurred during access to the peritoneum and trocar placement; the bladder was damaged on five occasions (12.5%), and 10 (25%) gastrointestinal injuries were reported. The ureter was involved in four ICs (10%). Vascular injuries accounted for 20% of the ICs in our series, while the obturator nerve was injured in three cases (7.5%). 

Furthermore, two conversions were observed: one to laparoscopic surgery during ePLND due to a robotic system malfunctioning and one to open surgery due to anesthesiologic adverse events. When we classified the ICs, two cases (5%) were classified as anesthesiologic ICs. Among the 38 surgical ICs, 16 (42%) were ePNLD-related, and 22 (58%) were RARP-related. ICs led to seven (0.37%) post-operative sequelae (four non-permanent and three permanent); the permanent sequelae were two obturator nerve impairments and one ureteral reimplantation due to stenosis development after DJ stent removal. No postoperative deaths due to intraoperative complications were reported in the present study.

Relative to ICs-naive patients, those with ICs were significantly older (67 years vs. 65 years, *p* = 0.02) and had a higher median BMI (27 vs. 26.1, *p* = 0.01), but did not differ in terms of comorbidities or tumor stage and characteristics (all *p* values ≥ 0.05; Table 2). 

Patients who had ICs did not have differences in terms of blood loss. However, the intraoperative transfusion rate was higher (13 vs. 0.9%, *p* < 0.01), and they had a higher rate of long lengths of stay (20 vs. 8%, *p* = 0.015).

## 4. Discussion

Due to the scarce availability of data regarding ICs during RARP, especially in the specific setting of RS-RARP, the current study provides a comprehensive review and detailed description of all ICs recorded within a contemporary series of more than 1800 PCa patients treated with RS-RARP at a single high-volume center. 

Robot-assisted radical prostatectomy (RARP) has emerged as a minimally invasive surgical technique for the treatment of prostate cancer. Prostate cancer is one of the most prevalent malignancies affecting men globally, and the quest for optimal treatment approaches has led to the integration of advanced technologies in surgical procedures. Compared to traditional open prostatectomy, RARP offers several advantages, including reduced blood loss, shorter hospital stays, and faster recovery times. The minimally invasive nature of RARP translates into smaller incisions, minimizing scarring and postoperative discomfort for patients. Additionally, the enhanced dexterity provided by robotic instruments allows surgeons to spare surrounding nerves critical for urinary and sexual function, reducing the risk of side effects commonly associated with prostate cancer surgery [1,2].

Despite all of the above advantages, RARP is not devoid of intraoperative complications (ICs), as with all the major surgical procedures that urologists perform every day, and intraoperative complications can lead to significant changes in surgical outcomes [19].

RS-RARP has emerged as a transformative technique in the management of localized PCa, offering patients the opportunity for curative treatment with acceptable morbidity [12]. RS-RARP has well-established advantages over the traditional RARP in terms of continence recovery [5,7], which has also been confirmed in challenging clinical presentations [20,21,22,23] and across different age groups [24]. 

This paper sheds light on the significant issue of ICs during the procedure. Our examination of common ICs during RS-RARP revealed several crucial insights. First and foremost, the prevalence of any grade of intraoperative complications remains relatively low (2.2%), highlighting the procedure’s overall safety when performed either by experienced or in-learning-curve surgeons, as previously demonstrated [25]. However, even though ICs are infrequent, they can have far-reaching consequences for both patients’ [26] and surgeons’ mental health [11]. When we analyze the intraoperative complications that cause short- or long-term side effects, the rate is very low: only 12 cases (0.6%) reported an adverse event with sequelae.

Patient-specific factors such as age, overall health status, and pre-existing medical conditions can significantly influence the likelihood of complications [12,15]. The operating team’s surgical expertise and experience also play a crucial role, as proficiency in robotic techniques can mitigate risks [27]. Prostate size, tumor location, and the presence of adhesions or scarring from prior surgeries further contribute to potential complications.

Hemorrhage, often due to vascular injuries during either trocar placement or lymph node dissection, remains a concern during RS-RARP. Proper training and surgical technique are pivotal in preventing excessive bleeding. Additionally, measures such as using energy devices and careful hemostasis can further reduce the risk of vascular injuries [15]. In our series, we reported five epigastric artery lesions during access or trocar placement, although these ICs occurred in the first years of our robotic experience. Vascular injuries during ePNDL are relatively common in our experience; however, they did not cause long-term consequences for the patients.

In oncological surgery, injuries to the obturator nerve may occur during pelvic lymph node dissection. While relatively uncommon, this complication is noteworthy, with reported rates in the literature reaching up to 0.7% of pelvic lymph node dissections (PLND). Depending on the shape and location of the injury, individuals may experience tingling and reduced sensation on the inner leg surface, coupled with the loss of motor function in the adductor muscles. Preventative measures include a thorough understanding of pelvic anatomy and the avoidance of electrocautery during lymph node dissection. In case of injury, prompt repair is recommended, focusing on a tension-free reapproximation of the nerve ends [28].

Ureteral injury, though less common, requires meticulous care and, when necessary, using ureteral stents to prevent strictures or leakages. When the injury is severe, ureteral reimplantation is needed to avoid stenosis and preserve renal function [28]. Special attention to anatomical landmarks and a thorough understanding of pelvic anatomy are essential during the posterior approach in isolating seminal vesicles and vasa deferentia. In almost 2000 cases, we reported only one ureteral injury during prostate isolation, against the common criticism that RS-RARP is prone to ureteral injuries.

Rectal injury, despite being rare, may lead to severe aftermaths. Vigilant dissection and adherence to anatomic landmarks in the RS-RARP technique are critical for avoiding rectal injuries. Additionally, advancements in imaging technologies can aid in identifying the precise location of critical structures during surgery [29]. In our series, we did not report rectal injuries despite the ten naive surgeons in learning curves. This fact could be due to a protective effect of RS-RARP. As the first plan to be developed, with instruments parallel to the rectal wall and not perpendicular, it is theoretically more difficult to cause rectal injuries.

The implications of these ICs are multiple. Prolonged surgical times, increased blood loss, and potential postoperative complications and sequelae can impact the overall patient postoperative course and recovery [15,17]. 

Patients treated by high-volume surgeons or in high-volume centers tend to experience fewer complications [27]. This emphasizes the importance of ongoing training and developing expertise in RARP.

Patient characteristics are a critical factor in mitigating ICs. Preoperative evaluation should consider factors such as prostate size, prior surgeries, radiations, and comorbidities, as these can influence the likelihood of complications [20,21,22,23]. Body mass index is particularly important, as obese patients with a BMI ≥ 30 kg/m^2^ had a higher risk of symptomatic lymphoceles after RARP [30]. Notably, we did not find any difference in oncological characteristics in patients who suffered complications and patients who did not. In addition, this study reports an older age and higher BMI in patients experiencing ICs, even if the relatively limited number of events prevented any assessment of independent predictors for ICs. 

This study is not devoid of limitations. First and foremost is the retrospective nature of the exploited data. Thus, our results must be interpreted within the bounds of the limitations of such data. Second, we reported a limited number of ICs, so it is not sufficient to conduct a multivariable analysis. Last, the experience reported may not be widely representative of daily clinical practice, since it is derived from a single high-volume center, which causes limited replicability and affects the results’ generalizability. 

## 5. Conclusions

Intraoperative complications during RS-RARP are relatively infrequent. However, they should not be underestimated. Patients suffering from ICs are older and have a higher body mass index, a higher rate of intraoperative blood transfusion, and a longer length of stay. Further prospective and retrospective studies should continue investigating strategies for minimizing intraoperative complications and optimizing patient outcomes during RS-RARP.

## Figures and Tables

**Table 1 cancers-16-01385-t001:** Overview of intraoperative complication cases.

Description of the Complication and EAUiaiC Grade		*n*, (%)	During RARP/ePNLD
Anesthesiologic ICs, *n* = 2 (5%)	Desaturation during pneumoperitoneum induction requiring open conversion Grade 2	1 (2.5)	
	Hemodynamic instability during ePLND, only one side performed Grade 4b	1 (2.5)	
Access and trocar placement ICs, *n* = 5 (12.5%)	Epigastric artery Injury Grade 1	5 (12.5)	RARP
Injury of intra-abdominal organs, *n* = 15 (37.5%)	Bladder injuries managed with immediate repair Grade 1	5 (12.5)	RARP
	Sigma injuries managed with immediate repair Grade 1	4 (10)	RARP
	Small bowel injuries requiring suture Grade 1	5 (12.5)	RARP
	Severe small bowel injury requiring resection and anastomosis Grade 2	1 (2.5)	RARP
Vascular injuries, *n* = 9 (22.5%)	Minor internal iliac artery injuries Grade 1	3 (7.5)	ePNLD
	Major internal iliac artery injury Grade 3	2 (5)	ePNLD
	Gluteal vein injuries Grade 1	2 (5)	ePNLD
	Iliac vein injuries Grade 1	2 (5)	ePNLD
Nerve, *n* = 3 (7.5%)	Obturatory nerve injury Grade 2	3 (7.5)	ePNLD
Ureteric injuries, *n* = 4 (10%)	Ureteral injury with suture and stenting Grade 2	2 (5)	ePNLD
	Ureteral injury with anastomosis/reimplantation Grade 4a	2 (5)	1 RARP 1ePNLD
Others, *n* = 2 (5%)	Needle loss Grade 1	1 (2.5)	RARP
	Robot malfunctioning Grade 4b	1 (2.5)	ePNLD

**Table 2 cancers-16-01385-t002:** Descriptive characteristics of 1891 prostate cancer patients treated with the Retzius-sparing approach, divided for intraoperative complications at a single European high-volume center.

	ICs (n = 40)	No ICs (n = 1851)	*p* Value
Age, years, median (IQR)	67 (62.9–73.3)	65 (59.9–69.2)	0.029
BMI, kg/mq, median (IQR)	27.0 (24.7–28.8)	26.1 (24.2–28.4)	0.018
Charlson comorbidity index, n (%)			0.9
0	27 (67.5)	1349 (72.9)	
1	5 (12.5)	235 (12.7)	
2	5 (12.5)	174 (9.4)	
3	3 (7.5)	93 (5.0)	
Previous abdominal surgery, n (%)	16 (240)	629 (34)	0.4
Previous surgery for BPH, n (%)	3 (7.5)	91 (4.9)	0.5
PSA at RS-RARP, ng/mL, median (IQR)	6.88 (4.7–11)	7.0 (5.1–9.9)	0.7
ISUP grade group at prostate biopsy, n (%)			0.1
1	15 (37.5)	859 (46.4)	
2	8 (20)	478 (25.8)	
3	4 (10)	194 (10.5)	
4	11 (27.5)	244 (13.2)	
5	2 (5)	76 (4.1)	
EAU risk classification group, n (%)			0.3
Low	12 (30)	614 (33.2)	
Intermediate	14 (35)	785 (42.4)	
High	14 (35)	452 (24.4)	
Estimated blood loss, ml, median (IQR)	250 (100–350)	200 (100–250)	0.2
Intraoperative blood transfusion, n (%)	3 (13.3)	17 (0.9)	<0.01
Length of stay > 3 days, n (%)	8 (20)	149 (8)	0.015

BMI: body mass index; BPH: benign prostate hyperplasia; PSA: prostate-specific antigen; ISUP: International Society of Urological Pathology; RS-RARP: Retzius-sparing robot-assisted radical prostatectomy; IQR: interquartile range; ISUP: International Society of Urological Pathology; EAU: European Association of Urology. The bold shows statistical significative variables.

## Data Availability

All data generated for this analysis were from an anonymized database. The code for the analyses will be made available upon request.

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
