# Peer review of "A Comprehensive Overview of Intraoperative Complications during Retzius-Sparing Robot-Assisted Radical Prostatectomy: Single Series from High-Volume Center"

_cancers, 2024, doi:10.3390/cancers16071385_

Round 1
Reviewer 1 Report
Comments and Suggestions for Authors
This is a relatively large, single-center study on intraoperative complications of RS-RARP. The Discussion provides specific details about complications that may occur with RS-RARP and serves as a reference for urologists when performing RS-RARP. Overall, the text is well-organized and easy to understand. However, perhaps due to the small number of events, the analysis was mediocre. Additionally, there did not seem to be much discussion regarding complications caused by the surgical technique of RS-RARP compared to conventional RARP. Couldn't the analysis be a little more ingenious? Considering that the readers of this paper are urologists who perform RARP, the content gave a somewhat unsatisfactory impression.
Comments on the Quality of English LanguageThe English language is easy to understand even for non-native speakers, and although it is written in simple sentences, it does not feel unrefined and I think it is relatively likable.
Author Response
Reviewer 1 This is a relatively large, single-center study on intraoperative complications of RS-RARP. The Discussion provides specific details about complications that may occur with RS-RARP and serves as a reference for urologists when performing RS-RARP. Overall, the text is well-organized and easy to understand. However, perhaps due to the small number of events, the analysis was mediocre. Additionally, there did not seem to be much discussion regarding complications caused by the surgical technique of RS-RARP compared to conventional RARP. Couldn't the analysis be a little more ingenious? Considering that the readers of this paper are urologists who perform RARP, the content gave a somewhat unsatisfactory impression. R: We thank the reviewer for the suggestion and the time spent. We improved the discussion accordingly to other reviewers. We included and addressed some of the common criticisms of the Retzius sparing approach regarding the risk of complications in the discussion (rectal and ureter injury). Making a proper comparison of INTRAOPERATIVE complications between different approaches would be very difficult because very few papers threaten ICS also during the traditional anterior approach. We are ready to extend further if needed.Reviewer 2 Report
Comments and Suggestions for Authors
The paper is well writen a presents intraoperative complications during RS RAPR
Please finds my comments below:
1. Is your study prospective or retropsecitve
I see opposite statements in the text
"This study is not devoid of limitations, first and foremost, the retrospective nature of
the exploited data"
2. Some significant informations are missing in the table 2. What about clinical T stage? Was advanced stage more commonly associated with complications?
3. Some more details should be provided regarding lenght of hospitalization, operative times, blood loss or transfusion.
4. "ICs led to seven (0.37%) post-operative sequelae (Four non-perma- 152
nent and three permanent)." what were the postoperative complications?
What was the sequele for complications? Did you record any re-operations/ postoperative deaths?
5. Did any patients received previous pelvis radiotherapy?
6. More discussion about the risk factors for complications should be mentioned.
7. The population that you have studied seems to be not representative for general population of prostate cancer patients. Please mention this and refer to other papers regarding the Charlson comorbidity index.
8. Please explain how is it possible that > 60% of pts in your study had Charlson index =0. How is it possible if age is also a factor in CCI assessment and patients > 50 yr have at least 1 point
Author Response
Reviewer 2 The paper is well writen a presents intraoperative complications during RS RAPR R: We thank the reviewer for the time spent and the precious comments. Please find my comments below: 1. Is your study prospective or retropsecitve I see opposite statements in the text "This study is not devoid of limitations, first and foremost, the retrospective nature of the exploited data" R: We apologise for the lack of clarity. While the database is maintained in a prospective way the study has been defined retrospectively. The text in the M&M have been modified accordingly. 2. Some significant informations are missing in the table 2. What about clinical T stage? Was advanced stage more commonly associated with complications? R: Thank you for the comment. Inside Table 2, oncological variables are included: PSA at RS-RARP—ISUP grade group at prostate biopsy, and EAU risk classification group that includes T stage. No correlation between stage and complication is found in our work, this have been specified in the main text. 3. Some more details should be provided regarding length of hospitalization, operative times, blood loss, or transfusion. R: We thank the reviewer for the suggestion. This has been added to the paper and surely improve our work. We did not include operative times because our series is a mix of RARP +/- LAD and could not be extrapolated. 4. "ICs led to seven (0.37%) postoperative sequelae (four non-permanent and three permanent)." What were the postoperative complications? What was the sequele for complications? Did you record any re-operations/ postoperative deaths? R: The point has been clarified in the result section. 5. Did any patients receive previous pelvis radiotherapy? R: No patients in the study received RT. 6. More discussion about the risk factors for complications should be mentioned. R: We thank the Reviewer for the suggestion, a paragraph has been added in the Discussion 7. The population that you have studied seems to be not representative for general population of prostate cancer patients. Please mention this and refer to other papers regarding the Charlson comorbidity index. 8. Please explain how is it possible that > 60% of pts in your study had Charlson index =0. How is it possible if age is also a factor in CCI assessment and patients > 50 yr have at least 1 point. 7 & 8 R: In the present paper, we used CCI, not age-adjusted, as the database was started in this way in 2010.Round 2
Reviewer 1 Report
Comments and Suggestions for Authors
Important additions have been made to the results and discussions, and I would like to appreciate this article as a comprehensive overview of RS-RARP.
Author Response
We thank the Reviewer for the good revision regarding our work and the time spent.
Reviewer 2 Report
Comments and Suggestions for Authors
Information that CCI is not age-adjusted was not provided despite previous comments.
Author Response
We thank the Reviewer for the excellent revision and the time spent.
The information requested has been added in M&M section.